# Comparison between Endoscopic Submucosal Dissection and Surgery in Patients with Early Gastric Cancer

**DOI:** 10.3390/cancers14153603

**Published:** 2022-07-24

**Authors:** Meng Qian, Yuan Sheng, Min Wu, Song Wang, Kaiguang Zhang

**Affiliations:** 1Department of Gastroenterology, The First Affiliated Hospital of USTC, Division of Life Sciences and Medicine, University of Science and Technology of China, Hefei 230001, China; mexni@outlook.com (M.Q.); syuan1009@163.com (Y.S.); 2Graduate School, Bengbu Medical College, Bengbu 233000, China; 3Department of Gastroenterology, Affiliated Provincial Hospital, Anhui Medical University, Hefei 230001, China; wumin8669@163.com

**Keywords:** early gastric cancer, endoscopic submucosal dissection, surgery, clinical outcomes

## Abstract

**Simple Summary:**

Conventional gastrectomy combined with regional lymph node dissection has been the standard treatment for early gastric cancer (EGC). This retrospective case–control study aimed to compare the clinical outcomes of endoscopic submucosal dissection (ESD) and surgical resection for EGC in China. After propensity score-matching, there were no significant differences between the two groups for OS, RFS, and DSS. Additionally, with similar R0 resection and recurrence rates, the ESD group showed less blood loss, fewer adverse events, lower hospital cost, and a shorter operative time and hospital duration than the surgery group. Therefore, ESD can be a first-line treatment of EGC in addition to surgery.

**Abstract:**

*Background*: Endoscopic submucosal dissection (ESD) has become a preferred treatment option for early gastric cancer (EGC). This study aimed to compare the clinical outcomes of ESD and surgical resection for EGC. *Methods*: This was a retrospective case–control study. Patients with a diagnosis of EGC who underwent ESD or surgery in our hospital from 2011 to 2020 were enrolled. We compared the clinical characteristics and treatment outcomes of these two groups according to propensity score-matching. The primary outcome comparison was overall survival (OS). Secondary outcomes were disease-specific survival (DSS), recurrence-free survival (RFS), and treatment-related events. *Results*: In the matched cohort, the ESD group showed comparable OS, RFS, and DSS with the surgery group. Statistical differences were shown in blood loss and adverse events. Furthermore, the ESD group showed lower hospital cost, as well as a shorter operative time and hospital duration than the surgery group. The R0 resection and recurrence rates were similar between the two groups. In Cox regression analysis, age, tumor size, poor differentiation, and lymphovascular invasion were regarded as independent factors of OS. *Conclusions*: With sufficient safety and advantages, ESD can be a first-line treatment of EGC. Preoperative evaluation is vital to the appropriate treatment and prognosis.

## 1. Introduction

As one of the most common malignancies worldwide, gastric cancer ranks fifth in terms of prevalence and fourth in terms of mortality [1]. Although the total number of people with gastric cancer has decreased in recent years, the incidence rate is increasing among young people [1]. In China, there are about 478,508 new cases of gastric cancer each year, making it the third leading cause of cancer-related deaths [2]. Therefore, choosing the appropriate treatment remains critical to improving clinical outcomes.

Early gastric cancer (EGC) is defined as a gastric cancer confined to the mucosal and submucosal layers, regardless of lymph node metastasis. Conventional gastrectomy combined with regional lymph node dissection is the standard treatment for EGC. In recent years, endoscopic submucosal dissection (ESD) has become an alternative treatment option for EGC due to its minimal invasiveness [3,4,5]. Many studies have shown comparable survival outcomes between patients receiving ESD versus surgery [6,7,8]. However, there have been some concerns of ESD. The greatest limitation is that ESD can only resect primary lesions, not lymph node metastases; thus, patients treated with ESD are considered to have a higher incidence of metachronous cancers [9,10,11]. Additionally, with the development of minimally invasive surgical techniques, laparoscopic gastrectomy has also shown favorable oncological efficacy [12,13]. Given these issues, the Japanese Gastroenterological Endoscopy Society (JGES), National Comprehensive Cancer Network (NCCN), British Society of Gastroenterology (BSG), and European Society of Gastrointestinal Endoscopy (ESGE) have instituted indications for endoscopic resection. The indication for ESD includes (1) all differentiated intramucosal carcinomas without ulceration, (2) differentiated intramucosal cancers ≤ 3 cm in diameter with ulceration, (3) undifferentiated intramucosal cancers ≤ 2 cm in diameter without ulceration, and (4) superficial submucosal invasive cancer (sm1 < 500 μm) of differentiated type without ulceration and tumor diameter ≤ 3 cm [14,15,16,17]. According to these guidelines, when determining the indications for ESD in the EGC, it depends significantly on the depth of infiltration and the risk of lymph node metastasis. Therefore, cancers that meet the indicated criteria tend to have a very low risk of lymph node metastasis [18]. With adequate preoperative evaluation, these lesions are considered to be curable by endoscopy.

In recent years, with advanced endoscopic technology, the detection of EGC has been significantly increased, and a better definition for the role of ESD in EGC management is needed. Although many previous studies have demonstrated the clinical value of ESD, large-scale studies comparing ESD with surgery are still lacking in China. Therefore, we conducted this study to evaluate the short- and long-term results between ESD and surgery in the treatment of patients with EGC.

## 2. Materials and Methods

### 2.1. Patients

A total of 2682 patients diagnosed with EGC who underwent ESD or surgery at The First Affiliated Hospital of USTC between January 2010 and December 2020 were initially enrolled. Exclusion criteria were as follows: (1) patients who lacked clinical information after diagnosis; (2) patients with a concurrent history of other gastrointestinal cancer; (3) conversion to surgical resection immediately in the ESD group; (4) adjuvant chemotherapy (neoadjuvant chemotherapy or radiation therapy) after surgery; (5) patients combined with serious diseases of other organs; (6) patients lost to follow-up. Finally, 531 ESD cases and 500 surgery cases were included in the study. The study protocol was approved by the Medical Ethics Committee of the First Affiliated Hospital of USTC (approval number: 2022-RE-051)

To minimize potential selection bias, 1:1 propensity score-matching (PSM) was performed on the basis of sex, age, cigarette, alcohol, family history, tumor location, size, infiltration depth, and the grade of tumor differentiation. Finally, we analyzed 274 patients in each group. The study flowchart is shown in Figure 1.

### 2.2. Data Collection

The baseline clinicopathologic characteristics and treatment outcomes were collected from the electronic medical record system, including age, sex, cigarette, alcohol, family history of gastric cancer, tumor characteristics (such as location, size, morphology, ulceration, depth of infiltration, grade of differentiation, and lymphovascular invasion), estimated blood loss, operative time, hospital duration, cost, resection margin, adverse events, and recurrence rate.

### 2.3. Treatment Procedures and Follow-Up

All patients in the study underwent an intensive preoperative evaluation, including gastroduodenoscopy, endoscopic ultrasound, and computed tomography. Patients and their families were informed by a clinician about the indications and details of ESD. The choice of treatment was based on the patient’s physical condition and preference, after explaining the advantages and disadvantages of ESD and surgery. The typical ESD procedure at our institution involved staining, marking, submucosal injection, mucosal incision, and submucosal dissection of the lesion. After staining the lesion with indigo carmine dye, an electrocautery marking was made 5 mm outside the margin with a Dual Knife (KD-650L, Olympus Medical Systems, Tokyo, Japan). Saline mixed with indigo carmine and epinephrine was injected into the submucosa to elevate the lesion. A circumferential incision was made along the outside of marked area, and the lesion was dissected with an insulated-tipped knife (IT knife, KD-611L, Olympus Medical Systems, Tokyo, Japan) or a Dual Knife (KD-650L, Olympus Medical Systems, Tokyo, Japan). During the procedure, a hemoclip (hemostatic clip) was used to control the bleeding. All patients in the surgery group underwent open or laparoscopy-assisted gastrectomy with D1 or D1+ lymph node dissection. The extent of gastric resection was determined by the tumor location, and lymph node dissection was performed according to the guidelines of Japanese Gastric Cancer [14,19,20].

Follow-up was recommended for patients who underwent ESD or surgery. In the ESD group, endoscopy was scheduled at 3, 6, and 12 months after initial ESD and then annually for 5 years. In the surgery group, endoscopic evaluation and abdominal CT were scheduled every 6 months for the first year and annually thereafter until 5 years. If patients were lost to follow-up, survival and recurrence information was obtained via telephone.

### 2.4. Definitions and Outcomes

Macroscopic types of tumor were classified according to the Japanese classification system, type I (protruding), type IIa (superficial elevated), type IIb (flat), type IIc (superficial depressed), and type III (excavated) [21]. We classified types I and IIa as elevated types, and types IIb, IIc, and III as flat or depressed types.

R0 resection was defined as resection with negative margins, in which there was no horizontal or vertical residual tumor at the resection margins.

Recurrences included metachronous, local, and regional or distant metastatic recurrence, while synchronous recurrence was not considered as recurrence. Synchronous and metachronous recurrences were defined as cancer detected at a site different from the primary lesion location within and after 12 months of ESD/surgery treatment, respectively. Local recurrence was defined as cancer detected at the scar site of ESD or at the anastomotic site of the surgical group. Regional/distant metastases were defined as the detection of a recurrence in other organs or lymph nodes.

The primary outcome comparison was overall survival (OS). Secondary outcomes were disease-specific survival (DSS), recurrence-free survival (RFS), and treatment-related events (including estimated blood loss, operative time, hospital duration, hospital cost, resection margin, adverse events, and recurrence rate). OS was defined as the time from ESD/surgery treatment to death of any cause, DSS was defined as the time from ESD/surgery treatment to death of gastric cancer, and RFS was defined as the time from ESD/surgery treatment to first recurrence or death of any cause.

### 2.5. Statistical Analysis

Continuous variables were presented as means ± standard deviations (SDs) or medians with interquartile ranges (IQRs), and categorical variables were presented as numbers with percentages. For comparisons, continuous variables were analyzed using the Student’s *t*-test or Mann–Whitney U-test, and categorical variables were analyzed using the chi-squared test or Fisher’s exact test. The Kaplan–Meier method was used to estimate survival analyses, and outcomes were compared using the log-rank test. Using the Cox proportional hazards regression model to perform univariate and multivariate analyzes, those variables that were considered clinically relevant or with *p* < 0.1 in univariate analysis were subsequently entered into multivariate analyses.

PSM was performed to minimize potential selection bias at a 1:1 ratio, with a match tolerance of 0.2 standard deviations of the logit of the estimated propensity score [22]. All statistical analyses were performed using SPSS software (version 26.0, IBM Corp, Armonk NY, USA) and Prism software (version 9.0, Inc., San Diego, CA, USA). A *p*-value <0.05 was considered statistically significant.

## 3. Results

### 3.1. Patient Characteristics

The baseline and clinicopathological characteristics of the study population before and after PSM are shown in Table 1. Before PSM, there were statistically significant differences in the family history of gastric cancer (*p* = 0.013), tumor location (*p* = 0.005), tumor size (*p* < 0.001), ulceration (*p* < 0.001), tumor differentiation grade (*p* < 0.001), and lymphovascular invasion (*p* = 0.010). After PSM, all characteristics were well balanced between the ESD and surgery groups (all *p* > 0.05).

### 3.2. Clinical Outcomes and Treatment-Related Events

The clinical outcomes before and after PSM are reported in Table 2. Patients in the ESD group had less blood loss, a shorter operative time and hospital duration, and a lower hospital cost (all *p* < 0.001). There was no statistically significant difference in the R0 resection rate and the recurrence rate. Additionally, the ESD group had significantly fewer adverse events compared with the surgery group (*p* < 0.001). In the matched population, these variables (including estimated blood loss, operative time, hospital duration, hospital cost, and adverse events) still showed statistically significant differences between the ESD and surgery groups (*p* < 0.001).

Adverse events that occurred in the ESD and surgery groups are summarized in Table 3 and Table 4, respectively. In the ESD group, one patient required emergency surgery due to massive bleeding, and one patient was transferred to surgery due to postoperative perforation. In the surgery group, two patients were transferred to the intensive care unit (ICU) due to severe infection. In terms of complications, there was no related death in the ESD group, while there was one related death in the surgery group (these cases are marked with an asterisk in the tables).

### 3.3. Survival Analysis

#### 3.3.1. Overall Survival

The Kaplan–Meier OS curves before and after matching are presented in Figure 2a. No significant difference in OS was found between the ESD and surgery groups (hazard ratio (HR) = 0.533, *p* = 0.060). After matching, there was also no significant difference in OS between the two groups (HR = 0.481, *p* = 0.065). The 5 year OS was comparable between the ESD and surgery groups (96.1% vs. 91.4%, *p* = 0.081).

For the PSM cohort, we further used Cox regression analysis to identify the variables associated with OS (Table 5). In the univariate analysis, age (*p* < 0.001), tumor size (*p* < 0.001), submucosal infiltration (*p* = 0.025), poor differentiation (*p* < 0.001), ulceration (*p* = 0.025), and the rate of R0 resection (*p* < 0.001) were associated with OS. In the multivariate analysis, there were still associations of age (HR = 1.24, 95% CI: 1.15–1.29, *p* < 0.001), tumor size (HR = 1.37, 95% CI: 1.11–1.69, *p* = 0.003), and poor differentiation (HR = 3.21, 95% CI: 1.13–9.10, *p* = 0.022). Meanwhile, treatment method (HR = 0.10, 95% CI: 0.04–0.26, *p* < 0.001) showed an association with OS.

#### 3.3.2. Recurrence-Free Survival

The Kaplan–Meier RFS curves before and after matching are presented in Figure 2b. There was no significant difference in RFS before (HR = 0.695, *p* = 0.241) and after (HR = 0.533, *p* = 0.103) matching between the ESD and surgery groups. The 5 year RFS in the ESD group of 95.8% was not significantly different from the 5 year RFS in the surgery group of 91.4% (*p* = 0.136).

The Cox regression model was used to analyze the factors contributing to RFS in the PSM cohort (Table 6). In the univariate analysis, factors associated with RFS included age (*p* < 0.001), tumor size (*p* < 0.001), poor differentiation (*p* = 0.001), and R1 resection (*p* = 0.001), while, in the multivariate analysis, age (HR = 1.24, 95% CI: 1.18–1.30, *p* < 0.001), tumor size (HR = 1.36, 95% CI: 1.07–1.73, *p* = 0.012), poor differentiation (HR = 3.41, 95% CI: 1.17–9.97, *p* = 0.025), and lymphovascular invasion (HR = 9.61, 95% CI: 3.21–28.79, *p* < 0.001) were the factors associated with RFS.

### 3.4. Disease-Specific Survival

The Kaplan–Meier DSS curves before and after matching are presented in Figure 2c. The DSS was significantly higher in the ESD group than in the surgery group (HR = 0.121, *p* < 0.001). However, it became comparable between the two groups in the matched cohort (HR = 0.312, *p* = 0.064). The 5 year DSS between the ESD and surgery groups was also not significantly different (98.6% vs. 95.7%, *p* = 0.062).

After matching, we also performed Cox regression analysis on the DSS (Table 7). The univariate analysis showed that age (*p* < 0.001), tumor size (*p* = 0.010), infiltration depth (*p* = 0.001), poor differentiation (*p* = 0.008), and ulceration (*p* = 0.027) were the factors for DSS, while multivariate analysis showed that treatment method (HR = 0.10, 95% CI: 0.02–0.46, *p* = 0.003), age (HR = 1.24, 95% CI: 1.15–1.33, *p* < 0.001), tumor size (HR = 1.38, 95% CI: 1.01–1.89, *p* = 0.041), and infiltration depth (HR = 5.75, 95% CI: 1.73–19.04, *p* = 0.004) were independent factors for DSS.

## 4. Discussion

In recent years, ESD has been developed as a treatment option for EGC that meets certain criteria [23]. Many studies have demonstrated the safety and efficacy of ESD in treating EGC. The outcomes of ESD treatment for gastric cancer included en bloc (92.4%), R0 (80.5%), curative resection rates (72.0%), complication (delayed bleeding: 5.9%, perforation: 4.2%), and recurrence rates (3 month: 2.3%; 12 month: 3.2%) [5]. A recent study reported favorable short- and long-term outcomes of endoscopic therapy in gastric cancer nationwide in Japan. In patients who met the absolute indications, the 5 year OS and DSS rates were 91.6% and 99.9%, respectively. In patients who belonged to expanded indications, the 5 year OS and DSS rates were still up to 90.3% and 99.7%, respectively [24]. Draganov et al. demonstrated the established outcomes of ESD for the treatment of gastrointestinal cancer. Regarding gastric cancer, the en bloc, R0, and curative resection rates were 98%, 82.2%, and 77.2%, respectively, and adverse events were reported in three cases of delayed bleeding (3%) and one case of perforation (1%) [25]. ESD is a safe and effective treatment option with favorable long-term results of early gastric cardiac cancer treated with ESD; en bloc, complete, and curative resection rates were 99.8%, 94.3%, and 80.5%, respectively. The 5 year OS rate was 89.6%, and the 5 year disease-specific survival rate reached 100% in patients with a curative resection [26].

However, there are still controversies regarding the oncological safety of ESD. For ESD specimens, R0 resection should be confirmed by pathological evaluation. A few patients who underwent ESD in our study needed additional surgery to get negative margins, but the rate was only 2.57% (26/1012). With its limitations, ESD can only remove the primary lesion, although the risk of lymph node metastasis in these EGCs is commonly low [27]. Compared with surgery, ESD has been reported to be associated with a higher incidence of metachronous tumors [28]. In addition, ESD requires a high level of professional skills, which limit its development in some areas. Due to factors such as epidemiological differences in gastric cancer, most reported results are from Asian regions, especially Korea and Japan. As one of the countries with a high prevalence of gastric cancer, China has contributed to nearly half of the gastric cancer-related deaths worldwide [29]. Although there have been some previous studies on comparing ESD with surgery in the treatment of EGC, reliable evidence from large-scale populations with long-term follow-up is still limited in China. We conducted this study to explore the role of ESD in patients with EGC in China.

Our results showed that blood loss in the ESD group was significantly less than that in the surgery group, while the operative time was significantly shorter in the ESD group. With the shorter duration of hospital stay, the cost of hospitalization was also significantly lower in the ESD group. These findings are consistent with previous studies [26,30]. Furthermore, a higher R0 resection rate of the ESD group was observed in the present study, which allowed ESD to show similar resection outcomes to surgery (99.4% vs. 100%, *p* = 0.092) [6,30]. Furthermore, the incidence of adverse events was significantly reduced in the ESD group compared with the surgery group [31]. From the results, we also observed that complications in the ESD group were milder than in the surgery group. Patients with complications in the ESD group could be successfully managed by conservative treatment or surgery, while some patients with severe complications in the surgery group needed to be transferred to the ICU, and even a case of treatment-related death was reported in the surgery group due to severe infection.

In the matched cohorts, patients in the ESD group showed comparable survival outcomes (including OS, RFS, and DSS) to that of the surgery group, along with additional advantages, such as shorter operative time and hospital duration, less blood loss, and lower treatment costs. Patients with gastric cancer who received ESD had a lower hospitalization cost, less traumatic experience, shorter recovery time, and better quality of life, while OS and DSS were not significantly different between the ESD and surgery groups [31]. Jeon et al. compared the long-term outcomes of ESD with surgery in the treatment of EGC. They concluded that the 5 year DSS rate of the ESD group was significantly better in the overall cohort, but similar to that of the surgery group in the matched cohort [32]. A study of young patients with gastric cancer further suggested that DSS was not constant, and that the prognostic factors that predict survival vary over time [33]. To verify the reliability of the results, we also compared the 5 year OS, RFS, and DSS between the ESD and surgery groups in the matched cohort, and we found that 5 year OS, RFS, and DSS were similar in the two populations. In this study, although only patients in the ESD group developed metachronous gastric cancer, we did not find a significant difference in the incidence of metachronous tumors between the two groups, and all of these cases were successfully treated with repeated ESD.

A multivariate analysis was used to identify the potential factors influencing OS, RFS, and DSS. For OS, age, tumor size, poor differentiation, and treatment method were independent factors. In a cohort matched study, Pyo et al. reported that the independent factors for OS included age, comorbidity index, performance index, sex, tumor morphology, and the depth of infiltration [34]. Regarding RFS, age, tumor size, poor differentiation, and lymphovascular invasion were influential factors. In terms of DSS, we found that age, tumor size, submucosal infiltration, and treatment method were negative prognostic factors. It is necessary to mention that age and tumor size were common risk factors. Tumor size has proven to be the independent risk factor for lymph node metastasis, significantly associated with the prognosis of patients with EGC [35,36,37]. Consequently, preoperative evaluation is crucial, and clinicians should comprehensively consider the relevant factors in choosing the appropriate treatment for patients with EGC.

For the treatment of EGC, ESD shows some advantages. First, ESD is more cost-effective, which can adequately save medical resources and reduce the burden of patients [3,38]. Second, ESD preserves the stomach integrity, avoiding potential functional complications relevant to surgery and, thus, improving the quality of life of patients [39]. Lastly, ESD provides comparable survival outcomes to surgery. Although ESD may have a higher incidence of metachronous lesions, successful treatment is available in most cases with repeated ESD, and a history of ESD does not negatively affect subsequent surgical treatment [40,41].

Our study had several limitations. Firstly, this was a single-center, retrospective study; hence, potential selection bias was unavoidable, although a PSM analysis was performed to reduce the clinicopathological differences between the two cohorts. Secondly, some patients were lost to follow-up during the study period. China is characterized by a large population and vast geographic area, and patients in the study were from all over the country, which increased the difficulty of follow-up. Due to the long timespan, some patients changed their phone numbers, and their follow-up information was not obtained. Fortunately, enough individuals were initially included in this study, and the sample was considered to be representative [42]. Therefore, we believe that the results were not significantly impacted. In the future, it is necessary to develop new strategies to improve the completeness and quality of follow-up information. Thirdly, the research time was 2 years later in the ESD group than in the surgery group; so as to minimize the effect of this limitation, we further compared the 5 year OS, RFS, and DSS. Lastly, the numbers for the matched cohort were limited; thus, we did not make subgroup comparisons based on the depth of infiltration, presence of ulceration, and lymphovascular invasion. Further multicenter and randomized studies are needed to present better evidence.

## 5. Conclusions

In conclusion, ESD provides comparable long-term OS, RFS, and DSS with surgery. Patients in the ESD group have significantly shorter hospital duration, lower hospital cost, and fewer adverse events. Thus, ESD is preferable to surgery for well-chosen localized gastric cancer when close follow-up is ensured. On the basis of the impact of tumor histological characteristics on survival outcomes, preoperative evaluation is essential for the selection of the appropriate treatment modality and patient prognosis.

## Figures and Tables

**Figure 1 cancers-14-03603-f001:**
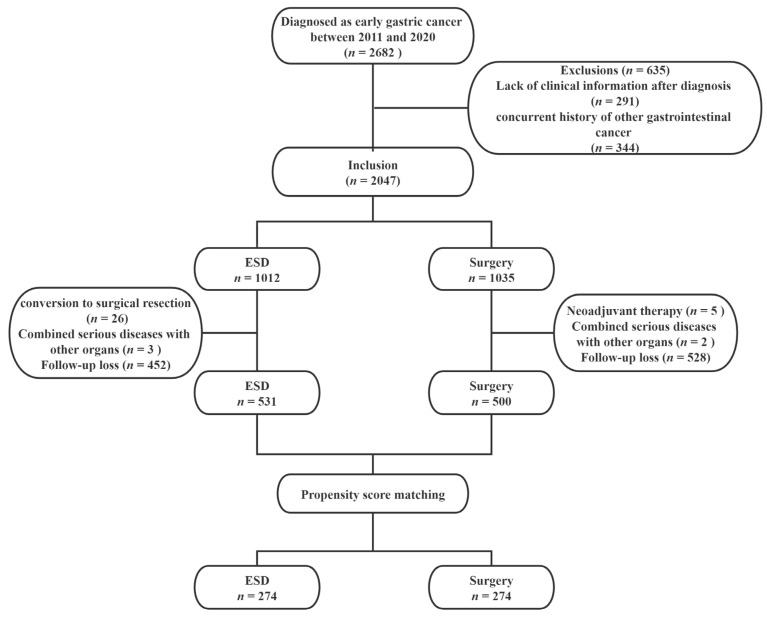
Flowchart of the study population. ESD: endoscopic submucosal dissection.

**Figure 2 cancers-14-03603-f002:**
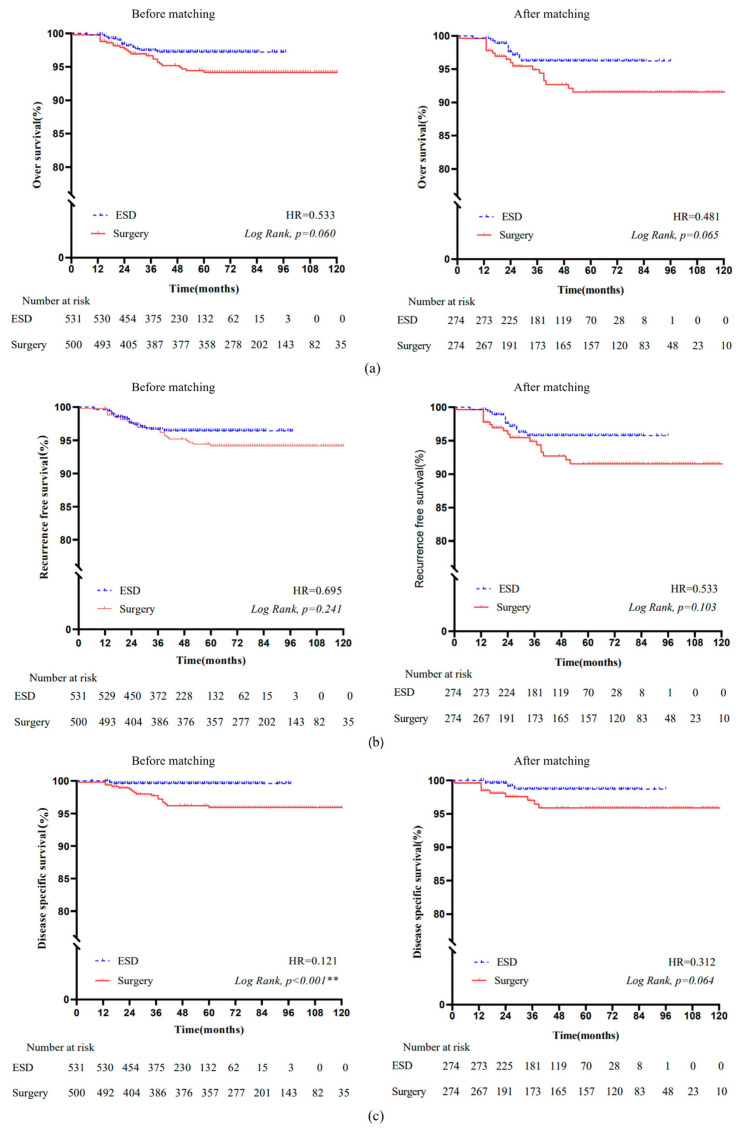
Kaplan–Meier survival curves of OS, RFS, and DSS in ESD and surgery groups. (**a**) OS before and after matching; (**b**) RFS before and after matching; (**c**) DSS before and after matching. Abbreviations: OS, overall survival; RFS, recurrence-free survival; DSS, disease-specific survival. **: *p* < 0.001.

**Table 1 cancers-14-03603-t001:** Baseline and clinicopathologic characteristics of the study population before matching After matching.

	ESD *N* = 531	Surgery *N* = 500	*p*-Value	ESD *N* = 274	Surgery *N* = 274	*p*-Value
Age (mean ± SD; years)	61.6 ± 10.7	61.4 ± 10.6	0.196	61.1 ± 10.7	61.6 ± 10.7	0.448
Sex, *n* (%)			0.506			0.851
Male	368 (69.3%)	356 (71.2%)		192 (70.1%)	194 (70.8%)	
Female	163 (30.7%)	144 (28.8%)		82 (29.9%)	80 (29.2%)	
Lifestyle, *n* (%)						
Cigarette	104 (19.6%)	108 (21.6%)	0.424	54 (19.7%)	53 (19.3%)	0.948
Alcohol Positive family history, *n* (%)	89 (16.8%) 22 (4.1%)	89 (17.8%) 39 (7.8%)	0.659 0.013	46 (16.8%) 13 (4.7%)	44 (16.1%) 21 (6.6%)	0.818 0.355
Tumor location, *n* (%)			0.005			0.390
Cardia	209 (39.4%)	143 (28.6%)		102 (37.2%)	94 (34.3%)	
Fundus	16 (3.0%)	13 (2.6%)		9 (3.3%)	8 (2.9%)	
Body	91 (17.1%)	100 (20.0%)		49 (17.9%)	54 (19.7%)	
Antrum	145 (27.3%)	154 (30.8%)		87 (31.7%)	77 (28.1%)	
Incisura angularis or pylorus	70 (13.2%)	90 (18.0%)		27 (9.9%)	41 (15.0%)	
Size (mean ± SD; cm)	2.6 ± 1.5	2.3 ± 1.1	<0.001 **	2.6 ± 1.5	2.4 ± 1.2	0.103
Tumor morphology, *n* (%)			0.932			0.913
Elevated	86 (16.2%)	80 (16.0%)		52 (19.0%)	51 (18.6%)	
Flat or depressed	445 (83.8%)	420 (84.0%)		222 (81.0%)	223 (81.4%)	
Tumor infiltration, *n* (%)			0.271			0.899
Mucosa	450 (84.7%)	411 (82.2%)		239 (87.2%)	238 (86.9%)	
Submucosa	81 (15.3%)	89 (17.8%)		35 (12.8%)	36 (13.1%)	
Ulceration, *n* (%)	32 (6.0%)	237 (47.4%)	<0.001 **	32 (11.7%)	48 (17.5%)	0.053
Tumor differentiation grade, *n* (%)			<0.001 **			0.062
Well-differentiated	363 (68.3%)	137 (15.3%)		122 (44.5%)	114 (41.6%)	
Moderately differentiated	156 (29.4%)	148 (15.3%)		140 (51.1%)	134 (48.9%)	
Poorly differentiated	12 (2.3%)	215 (15.3%)		12 (4.4%)	26 (9.5%)	
Lymphovascular invasion, *n* (%)	16 (3.0%)	32 (6.4%)	0.010	12 (4.4%)	13 (4.7%)	0.838

Abbreviations: ESD, endoscopic submucosal dissection; SD, standard deviation. **: *p* < 0.001.

**Table 2 cancers-14-03603-t002:** Clinical outcomes of the two study populations.

	ESD *N* = 531	Surgery *N* = 500	*p*-Value	ESD *N* = 274	Surgery *N* = 274	*p*-Value
Estimated blood loss (mL)			<0.001 **			<0.001 **
≤50	511 (96.2%)	68 (13.6%)		265 (96.7%)	50 (18.2%)	
>50	20 (3.8%)	432 (86.4%)		9 (3.6%)	224 (86.3%)	
Operative time (min), median (IQR)	73.5 (55.0–118.0)	195.5 (153.0–232.0)	<0.001 **	94.0 (60.0–120.0)	197.0 (153.0–232.0)	<0.001 **
Hospital duration (day), median (IQR)	9.0 (8.0–11.0)	17.0 (14.0–20.0)	<0.001 **	10.3 (8.0–11.0)	19.1 (14.0–21.0)	<0.001 **
Hospital cost (USD), median (IQR)	3121.1 (2693.1–3658.0)	6142.2 (5005.1–7211.7)	<0.001 **	3304.2 (2660.9–3438.9)	6507.6 (5157.7–7357.6)	<0.001 **
Resection margin			0.092			0.082
R0 resection	528 (99.4%)	500 (100.0%)		271 (98.9%)	274 (100.0%)	
R1 resection	3 (0.6%)	0 (0.0%)		3 (1.1%)	0 (0.0%)	
Recurrence	4 (0.8%)	1 (0.2%)	0.201	1 (0.4%)	0 (0.0%)	0.317
Adverse events, *n* (%)	7 (1.3%)	26 (5.2%)	<0.001 **	1 (0.4%)	16 (5.8%)	<0.001 **

Abbreviations: IQR, interquartile range. **: *p* < 0.001.

**Table 3 cancers-14-03603-t003:** Adverse events after ESD treatment for early gastric cancer patients.

Adverse Events	Total *N* = 7	Age	Sex	Location	Size (cm)	Infiltration	Estimated Blood Loss (mL)	Operative Time (min)	Hospital Duration (Day)	Hospital Cost (USD)
Gastropathy	2 (28.6%)	59	Male	Cardia	1.5	Mucosa	10	55	12	1203.9
		53	Male	Cardia	1.0	Mucosa	5	70	9	3162.0
Bleeding	4 (57.1%)	84	Female	Body	3.5	Mucosa	2	40	14	3964.1
		67	Male	Cardia	1.5	Mucosa	10	60	12	3861.5
		63 *	Male	Cardia	4.0	Mucosa	10	93	34	8450.7
		52	Male	Antrum	3.0	Mucosa	10	120	16	8708.5
Perforation	1 (14.3%)	54 *	Male	Cardia	4.5	Submucosa	10	140	12	5934.4

*: Transferred to surgery.

**Table 4 cancers-14-03603-t004:** Adverse events after surgery treatment for early gastric cancer patients.

Adverse Events	Total *N* = 26	Age	Sex	Location	Size (cm)	Infiltration	Estimated Blood Loss (mL)	Operative Time (min)	Hospital Duration (day)	Hospital Cost (USD)
Delayed gastric emptying	8 (30.9%)	55	Male	Antrum	2.0	Mucosa	200	214	39	12,373.8
		69	Female	Body	3.0	Submucosa	200	93	37	9534.3
		85	Male	Antrum	2.0	Mucosa	100	135	37	16,754.6
		64	Male	Incisura angularis	1.2	Mucosa	200	150	40	7747.8
		66	Female	Body	1.0	Submucosa	100	295	40	11,897.8
		54	Male	Incisura angularis	1.2	Submucosa	100	205	42	10,712.3
		68	Female	Antrum	1.0	Submucosa	50	125	51	9484.9
		63	Male	Body	3.0	Submucosa	100	246	14	5390.5
Anastomotic leakage	3 (11.5%)	63	Male	Body	5.0	Submucosa	100	150	54	8917.2
		71	Female	Cardia	3.5	Submucosa	200	317	36	9378.0
		60	Male	Antrum	1.5	Mucosa	100	200	41	8457.5
Anastomotic leakage + gastroparesis	1 (3.8%)	70	Female	Body	2.0	Mucosa	200	140	99	13,457.5
Ileus	4 (15.4%)	61	Male	Antrum	3.0	Submucosa	100	265	47	13,814.6
		37	Male	Antrum	1.5	Mucosa	100	180	23	5488.9
		78	Female	Antrum	8.0	Mucosa	100	215	25	7170.1
		67	Male	Incisura angularis	2.0	Submucosa	100	205	14	7371.2
Bleeding	2 (7.7%)	63	Male	Body	2.5	Mucosa	100	154	44	6542.9
		79	Male	Cardia	2.5	Submucosa	100	180	47	7887.4
Infection	3 (11.5%)	69	Male	Incisura angularis	1.0	Submucosa	100	220	42	6533.3
Severe infection **		78	Male	Antrum	5.0	Mucosa	100	155	28	21,170.6
Severe infection *		60	Male	Antrum	2.0	Submucosa	300	235	25	21,101.8
Fever	4 (15.4%)	72	Female	Cardia	1.5	Submucosa	100	187	29	11,168.6
		59	Male	Cardia	1.5	Mucosa	100	190	20	8965.7
		67	Male	Cardia	2.0	Mucosa	200	189	46	11,228.6
		70	Female	Cardia	2.5	Submucosa	200	305	29	9630.1
Other	1 (3.8%)	86	Male	Antrum	2.0	Mucosa	200	162	28	9204.0

**: Transferred to intensive care unit and then died; *: Transferred to intensive care unit.

**Table 5 cancers-14-03603-t005:** Univariate and multivariate regression analyses of overall survival for propensity score-matching patients.

Variables	Univariate Analysis		Multivariate Analysis	
	Hazard Ratio (95% CI)	*p*-Value	Hazard Ratio (95% CI)	*p*-Value
Treatment method (ESD vs. surgery)	0.48 (0.22–1.07)	0.072	0.10 (0.04–0.26)	<0.001 **
Age	1.20 (1.15–1.25)	<0.001 **	1.24 (1.15–1.29)	<0.001 **
Sex, male = 1	0.31 (0.09–1.03)	0.055		
Cigarette, no = 1	1.60 (0.55–4.64)	0.384		
Alcohol, no = 1	2.64 (0.63–11.15)	0.286		
Positive family history	22.01 (0.02–26.86)	0.394		
Tumor location		0.593		
Cardia	1.00	_		
Fundus	0.32 (0.07–1.44)	_		
Body	1.04 (0.44–2.47)	_		
Antrum	1.24 (0.44–3.53)	_		
Incisura angularis or pylorus	0.02 (0.27–1.54)	_		
Tumor size	1.49 (1.22–1.82)	<0.001 **	1.37 (1.11–1.69)	0.003
Tumor morphology		0.505		
Elevated	1.00	_		
Flat or depressed	1.44 (0.50–4.15)	_		
Submucosal infiltration	2.47 (1.12–1.81)	0.025		
Tumor differentiation grade				
Well-differentiated	1.00	_		
Moderately differentiated	1.68 (0.69–4.11)	0.257		
Poorly differentiated	5.02 (1.88–13.41)	0.001	3.21 (1.13–9.10)	0.022
Ulceration, no = 1	2.50 (0.18–0.89)	0.025		
Lymphovascular invasion, no = 1	1.97 (0.89–4.40)	0.097		
Resection margin				
R0 resection	1.00	_		
R1 resection	5.88 (0.06–1.47)	0.001		

**: *p* < 0.001.

**Table 6 cancers-14-03603-t006:** Univariate and multivariate regression analyses of recurrence-free survival for propensity score-matching patients.

Variables	Univariate Analysis		Multivariate Analysis	
	Hazard Ratio (95% CI)	*p*-Value	Hazard Ratio (95% CI)	*p*-Value
Treatment method (ESD vs. surgery)	0.53 (0.25–1.16)	0.111		
Age	1.18 (1.13–1.25)	<0.001 **	1.24 (1.18–1.30)	<0.001 **
Sex, male = 1	0.30 (0.09–0.98)	0.056		
Cigarette, no = 1	1.27 (0.48–3.35)	0.624		
Alcohol, no = 1	2.75 (0.65–11.57)	0.169		
Positive family history	22.02 (0.02–23.49)	0.385		
Tumor location		0.602		
Cardia	1.00	_		
Fundus	0.43 (0.25–1.56)	_		
Body	0.32 (0.07–1.44)	_		
Antrum	0.92 (0.38–2.26)	_		
Incisura angularis or pylorus	1.24 (0.44–3.53)	_		
Tumor size	1.52 (1.24–1.85)	<0.001 **	1.36 (1.07–1.73)	0.012
Tumor morphology		0.853		
Elevated	1.00	_		
Flat or depressed	1.40 (0.42–2.99)	_		
Submucosal infiltration	2.20 (1.96–5.04)	0.061		
Tumor differentiation grade				
Well-differentiated	1.00	_		
Moderately differentiated	1.92 (0.76–4.88)	0.170		
Poorly differentiated	5.72 (2.07–15.79)	0.001	3.41 (1.17–9.97)	0.025
Ulceration, no = 1	2.17 (0.20–1.04)	0.062		
Lymphovascular invasion, no = 1	2.21 (0.96–5.09)	0.063	9.61 (3.21–28.79)	<0.001 **
Resection margin				
R0 resection	1.00	_		
R1 resection	5.88 (0.06–0.50)	0.001		

**: *p* < 0.001.

**Table 7 cancers-14-03603-t007:** Univariate and multivariate regression analyses of disease-specific survival for propensity score-matching patients.

Variables	Univariate Analysis		Multivariate Analysis	
	Hazard Ratio (95% CI)	*p* Value	Hazard Ratio (95% CI)	*p*-Value
Treatment method (ESD vs. surgery)	0.31 (0.08–1.15)	0.080	0.10 (0.02–0.46)	0.003
Age	1.19 (1.11–1.27)	<0.001 **	1.24 (1.15–1.33)	<0.001 **
Sex, male = 1	0.22 (0.03–1.73)	0.151		
Cigarette, no = 1	2.99 (0.39–23.24)	0.293		
Alcohol, no = 1	2.28 (0.57–3.41)	0.431		
Positive family history	21.99 (0.10–9.54)	0.572		
Tumor location		0.951		
Cardia	1.00	_		
Fundus	0.01 (0.02–0.54)	_		
Body	0.37 (0.13–1.24)	_		
Antrum	0.77 (0.22–2.62)	_		
Incisura angularis or pylorus	0.43 (0.05–3.47)	_		
Tumor size	1.48 (1.10–2.00)	0.010	1.38 (1.01–1.89)	0.041
Tumor morphology		0.649		
Elevated	1.00	_		
Flat or depressed	0.74 (0.20–2.73)	_		
Submucosal infiltration	7.36 (2.33–23.21)	0.001	5.75 (1.73–19.04)	0.004
Tumor differentiation grade				
Well-differentiated	1.00			
Moderately differentiated	7.82 (0.96–6.61)	0.054		
Poorly differentiated	19.40 (0.96–6.61)	0.008		
Ulceration, no = 1	0.27 (0.09–0.86)	0.027		
Lymphovascular invasion, no = 1	1.53 (0.41–5.67)	0.144		
Resection margin				
R0 resection	1.00	_		
R1 resection	3.22 (0.04–2.44)	0.269		

**: *p* < 0.001.

## Data Availability

Deidentified individual participant data are available and can be provided on reasonable request to the corresponding author.

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
