# Peer review of "Comparison between Endoscopic Submucosal Dissection and Surgery in Patients with Early Gastric Cancer"

_cancers, 2022, doi:10.3390/cancers14153603_

Round 1

Reviewer 1 Report

I have no additional questions for authors. 

Author Response

Comments: I have no additional questions for authors.

Response: Thank you for taking the time to read our paper, and we gratefully appreciate the positive comments on our work.

Reviewer 2 Report

The study is interesting and well designed. It could be a solid basis for a prospective study. The interest for EGC is increasing worldwide, not only in eastern countries, and the results of ESD as a treatment are very interesting. More solid prospective studies are needed, to investigate the risk factors of lymph node metastasis even in such early stage of disease.

Author Response

Comments: The study is interesting and well designed. It could be a solid basis for a prospective study. The interest for EGC is increasing worldwide, not only in eastern countries, and the results of ESD as a treatment are very interesting. More solid prospective studies are needed, to investigate the risk factors of lymph node metastasis even in such early stage of disease.

Response: Thank you for reading our paper carefully and giving the positive comments on our work. We strongly agree with your opinion. The investigate of the risk factors for lymph node metastasis is very significant for the selection of reasonable treatment and improvement of prognosis. In the future, we are also interested in performing prospective study to explore the risk factors for lymph node metastasis.

Reviewer 3 Report

This study compared the oncological outcomes between ESD and surgery. I have some suggestions listed below. 

1. As you mentioned in the Methods section, endoscopy and image study were performed for every included patients. I wonder if the patients received endoscopic ultrasound to assess the depth of tumor invasion. 

2. As the data in Table 1, poor differentiated gastric cancer was more common in the surgery group than in the ESD group. In my experiences, diffuse type gastric cancer had a higher proportion of poor differentiated histology, which was relative contra-indication for ESD. How about the distribution of final pathology (cancer staging and histology) between two groups before and after natching? 

Author Response

We would like to thank you for your careful reading, helpful comments, and constructive suggestions, which are valuable for improving our paper. Your copious experience and professional insights into the disease make us respectful. We have considered your suggestions carefully and revised our manuscript according to it.

Comment 1: As you mentioned in the Methods section, endoscopy and image study were performed for every included patients. I wonder if the patients received endoscopic ultrasound to assess the depth of tumor invasion.

Response 1: Thank you for your suggestions. In our hospital, every included patients also received endoscopic ultrasound to assess the depth of tumor invasion. The role of endoscopic ultrasound in establishing the feasibility of endoscopic resection is somewhat controversial [1-3]. It was reported to have a relatively limited diagnostic accuracy, so we didn't mention this in our article. According to your suggestions, we have added it to the revised manuscript (in the Methods section, page 3, line 101-102).

[1] Pimentel-Nunes, P.; Libânio, D.; Bastiaansen, B.A.J.; Bhandari, P.; Bisschops, R.; Bourke, M.J.; Esposito, G.; Lemmers, A.; Maselli, R.; Messmann, H.; et al. Endoscopic submucosal dissection for superficial gastrointestinal lesions: European Society of Gastrointestinal Endoscopy (ESGE) Guideline - Update 2022. Endoscopy 2022, 54, 591-622, doi:10.1055/a-1811-7025.

[2] Ajani, J.A.; D'Amico, T.A.; Bentrem, D.J.; Chao, J.; Cooke, D.; Corvera, C.; Das, P.; Enzinger, P.C.; Enzler, T.; Fanta, P.; et al. Gastric Cancer, Version 2.2022, NCCN Clinical Practice Guidelines in Oncology. J Natl Compr Canc Netw 2022, 20, 167-192, doi:10.6004/jnccn.2022.0008.

[3] Pimentel-Nunes, P.; Dinis-Ribeiro, M.; Ponchon, T.; Repici, A.; Vieth, M.; De Ceglie, A.; Amato, A.; Berr, F.; Bhandari, P.; Bialek, A.; et al. Endoscopic submucosal dissection: European Society of Gastrointestinal Endoscopy (ESGE) Guideline. Endoscopy 2015, 47, 829-854, doi:10.1055/s-0034-1392882.

Comment 2: As the data in Table 1, poor differentiated gastric cancer was more common in the surgery group than in the ESD group. In my experiences, diffuse type gastric cancer had a higher proportion of poor differentiated histology, which was relative contra-indication for ESD. How about the distribution of final pathology (cancer staging and histology) between two groups before and after matching?

Response 2: We agree with your comments. Compare with intestinal gastric cancer, diffuse gastric cancer tends to have a higher proportion of poor differentiated histology, which was relative contra-indication for ESD. The expanded indications for ESD in early gastric cancer include: 1) intramucosal tumor, differentiated type, without ulcerative findings, and > 2 cm in size; 2) intramucosal tumor, differentiated type, with ulcerative findings, and ≤ 3 cm in size; 3) intramucosal tumor, undifferentiated type, without ulcerative findings, and ≤ 2 cm in size [4,5]. After explaining the advantages and disadvantages of ESD and surgery, most patients with poor differentiated gastric cancer opted for surgery. Therefore, poor differentiated gastric cancer was more common in the surgery group than in the ESD group before matching. To minimize potential selection bias, propensity score-matching was peformed in our study. After matching, histologic grade was well-balanced between the two groups. The tumer infiltration depth and differentiation grade in this study were determined according to histological examination of the specimens [5], and the distribution of final pathology between two groups before and after matching were showed in Table 1.

[4] Japanese gastric cancer treatment guidelines 2018 (5th edition). Gastric Cancer 2021, 24, 1-21, doi:10.1007/s10120-020-01042-y.

[5] National Health Commission Of The People's Republic Of, C. Chinese guidelines for diagnosis and treatment of gastric cancer 2018 (English version). Chin J Cancer Res 2019, 31, 707-737, doi:10.21147/j.issn.1000-9604.2019.05.01.

Round 2

Reviewer 3 Report

1. In your study population, there was around 15% patients with submucosal invasion. Would you do further surgical resection in these cases? 

2. The analysis done by Kaplan-Meier survival curves is unadjusted analysis. Would you please include intervention modality (ESD versus surgical resection) as a variable into adjusted analusis models? With this data, it is more reasonable to draw the conclusion. 

Author Response

Comment 1: In your study population, there was around 15% patients with submucosal invasion. Would you do further surgical resection in these cases?

Response 1: Thank you for your comments. Submucosal invasion depth of  ≥500 μm was significantly associated with lymph node metastasis. For these patients, we suggested to receive additional surgery, and follow-up methods according to different eCura evaluation levels [1,2]. The risk of lymph node metastasis and possibility of the subsequent local recurrence and/or distant metastasis were assessed and explained sufficiently to the patients. The final decision for whether to perform additional surgery in these cases was based on the patient's adequately informed consent.

[1] Japanese gastric cancer treatment guidelines 2018 (5th edition). Gastric Cancer 2021, 24, 1-21, doi:10.1007/s10120-020-01042-y.

[2] National Health Commission Of The People's Republic Of, C. Chinese guidelines for diagnosis and treatment of gastric cancer 2018 (English version). Chin J Cancer Res 2019, 31, 707-737, doi:10.21147/j.issn.1000-9604.2019.05.01.

Comment 2: The analysis done by Kaplan-Meier survival curves is unadjusted analysis. Would you please include intervention modality (ESD versus surgical resection) as a variable into adjusted analusis models? With this data, it is more reasonable to draw the conclusion.

Response 2: We are grateful for the suggestion. To make the conclusion more reasonable, we have added intervention modality (ESD versus surgical resection) as a variable into adjusted analysis models (in page 8-11, Table 5, 6, and 7). Variables that were considered clinically relevant or with P < 0.1 in univariate were subsequently entered into multivariate analyzes. When the cox analysis was re-performed, the results also changed. For overall survival, risk factors adjusted from “age (HR=1.24, 95% CI: 1.18-1.29, p<0.001), tumor size (HR=1.32, 95% CI: 1.04-1.86, p=0.021), poorly differentiation (HR=3.21, 95% CI: 1.13-9.10, p=0.028), and lymphovascular invasion (HR=7.90, 95% CI: 2.80-8.27, p<0.001)” to “age (HR=1.24, 95% CI: 1.15-1.29, p<0.001), tumor size (HR=1.37, 95% CI: 1.11-1.69, p=003), poorly differentiation (HR=3.21, 95% CI: 1.13-9.10, p=0.022), and treatment method (HR=0.10, 95% CI: 0.04-0.26, p<0.001)” (page 8, line 201-204). For recurrence-free survival, there was no change in the results of the multivariate analysis. For disease-specific survival, independent factors changed from “age (HR=1.23, 95% CI: 1.14-1.33, p<0.001), tumor size (HR=1.39, 95% CI: 1.02-1.89, p=0.039), infiltration depth (HR=5.94, 95% CI: 1.79-19.67, p=0.004), and lymphovascular invasion (HR=9.28, 95% CI: 2.01-42.84, p=0.003)” to “treatment method (HR=0.10, 95% CI: 0.02-0.46, p=0.003), age (HR=1.24, 95% CI: 1.15-1.33, p<0.001), tumor size (HR=1.38, 95% CI: 1.01-1.89, p=0.041), and infiltration depth (HR=5.75, 95% CI: 1.73-19.04, p=0.004)” (page 10, line 233-236). Following the results, we changed our discussion section accordingly. (page 13, line 333-334 and line 338-341)

Round 3

Reviewer 3 Report

My comments had been addressed comprehensively. 

This manuscript is a resubmission of an earlier submission. The following is a list of the peer review reports and author responses from that submission.

Round 1

Reviewer 1 Report

There is a fatal problem with patient selection and matching. For these patient group, differentiation and depth of invasion are the most important factors to be considered in patient selection.

Reviewer 2 Report

I'm agree to publish the paper.

Reviewer 3 Report

We thank the authors for their contribution in the diagnosis, evaluation and treatment (surgical and  endoscopic ) in patients with early gastric cancer.

The paper is well structured and the study numbers are significant. The endoscopic treatment of early gastric cancer can be an excellent treatment opportunity as an alternative to surgery, where there are physician who can perform this procedure  especially in older patients with high anesthetic risks. Obviously these results depend a lot on the pre-treatment diagnosis and staging

So I believe that the article is important from a scientific point of view and that it can be published.

Reviewer 4 Report

The article approaches a very interesting topic and addresses an issue that is under debate as more and more mininally invasive procedures are gaining weight against major surgery which was the norm in the past. 

Endoscopic submucosal resection may be a notable alternative in selected cases of early gastric cancer. 

The introduction provides enough information on this topic and mentions current guidelines from Japan, UK, Europe and US (NCCN). 

The authors accurately describe the methods used to assess the effectiveness of ESD in Chinese patients, on an adequate sample of patients and propensity score-matching is very welcome in this comparison. 

The article is valuable, but as the authors mentioned, it would be important to see the impact of lymph node and vascular invasion to establish whether ESD is a contender in more aggresive types of cancer. Please address this issue, by further explaining why it was not evaluated in a separate subgroup. 

A good paper overall and I have no further comments.